# Incidence and Outcomes of Pulmonary Embolism among Hospitalized COVID-19 Patients

**DOI:** 10.3390/ijerph18147645

**Published:** 2021-07-18

**Authors:** Omaima Ibrahim Badr, Hassan Alwafi, Wael Aly Elrefaey, Abdallah Y. Naser, Mohammed Shabrawishi, Zahra Alsairafi, Fatemah M. Alsaleh

**Affiliations:** 1Department of Chest Medicine, Faculty of Medicine, Mansoura University, Mansoura 35516, Egypt; omaimabadr78@yahoo.com; 2Department of Pulmonary Medicine, Al Noor Specialist Hospital, Mecca 20424, Saudi Arabia; walrefaei@moh.gov.sa (W.A.E.); shabrawishi.m@gmail.com (M.S.); 3Faculty of Medicine, Umm Al Qura University, Mecca 21514, Saudi Arabia; hhwafi@uqu.edu.sa; 4Department of Applied Pharmaceutical Sciences and Clinical Pharmacy, Faculty of Pharmacy, Isra University, Amman 11622, Jordan; abdallah.naser@iu.edu.jo; 5Department of Internal Medicine, Al Noor Specialist Hospital, Mecca 20424, Saudi Arabia; 6Department of Pharmacy Practice, Faculty of Pharmacy, Kuwait University, Kuwait City 12037, Kuwait; zahra.alsairafi@ku.edu.kw

**Keywords:** COVID-19, hospitalized, pulmonary embolism, survival

## Abstract

Objectives: Patients with COVID-19 may be at high risk for thrombotic complications due to excess inflammatory response and stasis of blood flow. This study aims to assess the incidence of pulmonary embolism among hospitalized patients with COVID-19, risk factors, and the impact on survival. Methods: A retrospective case-control study was conducted at Al-Noor Specialist Hospital in Saudi Arabia between 15 March 2020 and 15 June 2020. Patients with confirmed COVID-19 diagnosis by a real-time polymerase chain reaction (PCR) and confirmed diagnosis of pulmonary embolism by Computed Tomography pulmonary angiogram (CTPA) formed the case group. Patients with confirmed COVID-19 diagnosis by a real-time polymerase chain reaction (PCR) and without confirmed diagnose of pulmonary embolism formed the control group. Logistic regression analysis was used to identify predictors of pulmonary embolism and survival. Results: A total of 159 patients participated were included in the study, of which 51 were the cases (patients with pulmonary embolism) and 108 patients formed the control group (patients without pulmonary embolism). The incidence of PE among those hospitalized was around 32%. Smoking history, low level of oxygen saturation, and higher D-dimer values were important risk factors that were associated with a higher risk of developing PE (*p* < 0.05). Higher respiratory rate was associated with higher odds of death, and decreased the possibility of survival among hospitalized patients with PE. Conclusions: Pulmonary embolism is common among hospitalized patients with COVID-19. Preventive measures should be considered for hospitalized patients with smoking history, low level of oxygen saturation, high D-dimer values, and high respiratory rate.

## 1. Introduction

The novel coronavirus, severe acute respiratory syndrome coronavirus 2 (SARS-CoV-2), emerged in Wuhan city, China, at the end of 2019, and since then, it has spread to more than 190 countries across the world. The total number of confirmed cases as of August 2020 was >20 million, with >700,000 confirmed deaths [1]. The severity of the disease can range from being asymptomatic to mild, moderate, or severe, with multi-organ failure and death [2,3]. Acute respiratory distress syndrome, which is one of the major complications of the coronavirus disease of 2019 (COVID-19), is associated with a high mortality rate and is considered the main cause of death [4,5]. Venous and arterial thrombosis is considered one of the most severe consequences of the disease and has poor prognostic outcomes [6,7]. Patients with COVID-19 may be at a high risk of thrombotic complications due to excess inflammatory response, platelet activation, endothelial dysfunction, and stasis of blood flow [8,9]. At present, it has been suggested that patients with COVID-19 infection have an increased risk of thrombosis [10]. In addition, recent data have suggested that increased levels of the D-dimer test can be a predictor of adverse outcomes such as underlying coagulopathy and thrombotic risk in patients with COVID-19 [11,12]. However, the published data are very limited, with no previous studies on the incidence and risk factors of PE in patients with COVID-19 in the Middle East. Therefore, this study aims to assess the incidence of pulmonary embolism among hospitalized patients with COVID-19, risk factors, and the impact on survival.

## 2. Methods

### 2.1. Study Design

A retrospective case-control study was conducted at Al-Noor Specialist Hospital in Mecca, Saudi Arabia. The description of the study settings and the hospital has been described previously [1].

### 2.2. Inclusion and Exclusion Criteria

All patients had a confirmed COVID-19 diagnosis by a real-time polymerase chain reaction (PCR). The PCR samples were obtained through a nasopharyngeal swap. The patients underwent a Computed Tomography pulmonary angiogram (CTPA) for the diagnosis of pulmonary embolism (PE). All patients were admitted between 15 March 2020, and 15 June 2020; they were followed up for a time to assess the clinical outcome, and the final date of follow-up was 15 August 2020. Patients younger than 15 years old, and patients who were not eligible for CTPA pulmonary angiography, were excluded from the study.

### 2.3. Data Collection and Study Variables

Data were collected from medical files and electronic records using a unique medical record number (MRN) for each patient. Data included the patient’s demographics, clinical symptoms (fever, cough, SOB, nausea, vomiting, diarrhea, headache, body aches, loss of smell, and loss of taste), comorbidities, and chest radiograph. These data were collected on admission to the hospital. Clinical signs, including heart rate, respiratory rate, and SO2% on room air, were collected at the time of CTPA. Data regarding risk assessments for thromboembolism including age, smoking, obesity (defined as BMI > 30), D-dimer level, the severity of the disease (critical (in ICU) and non-critical (outside ICU)), the time between the admission and the CTPA, the presence of comorbidities, the risk of disseminated intravascular coagulation (DIC) including CBC (WBC, platelets, and hemoglobin level), and the coagulation parameters (PT, PTT, and INR) were collected in the same day or within 24 h of the Computed Tomography pulmonary angiogram (CTPA). The radiological examinations including the CTPA were reviewed by a senior certified chest radiologist that classified the pulmonary embolism to the main trunk, lobar, segmental, and subsegmental; RV strain by CT was defined as a right ventricle to left ventricle size ratio of ≥0.9. In addition, pulmonary systolic pressure, right ventricular (RV) dysfunction (diagnosed by qualitative and quantitative RV dilation or RV systolic dysfunction), left ventricular (LV) dysfunction (defined as decreased left ventricular systolic function with decreased LV ejection fraction, regional wall motion abnormalities, or both), and visualization of thrombus in the right heart or the pulmonary artery were diagnosed through trans-thoracic echocardiography, which was reviewed by an experienced, board-certified echocardiography attending physician.

### 2.4. Case Definition

Patients with a confirmed COVID-19 diagnosis by a real-time polymerase chain reaction (PCR) and with a confirmed diagnosis of pulmonary embolism formed the case group.

### 2.5. Control Definition

Patients with a confirmed COVID-19 diagnosis by a real-time polymerase chain reaction (PCR) and without a confirmed diagnosis of pulmonary embolism formed the control group.

### 2.6. Outcomes

The outcome predictors of those patients were admission to an intensive care unit, intubation and connection to mechanical ventilation, thrombolytic therapy or catheter-directed thrombolysis, bleeding (minor or major bleeding) as a complication of PE therapy, a prolonged hospital stay (>2 weeks), and mortality.

### 2.7. Statistical Analysis

Descriptive statistics were used to describe patients’ demographic characteristics, radiological findings, clinical signs and symptoms, and comorbidities. Continuous data were reported as mean ± SD for normally distributed data and as median (interquartile range (IQR)) for not normally distributed data, and categorical data were reported as percentages (frequencies). Independent sample *t*-test/ANOVA was used to compare the mean value for normally distributed continuous variables and The Mann–Whitney U test/Kruskal–Wallis test was used to compare the median value for not normally distributed ones. A Chi-squared test/Fisher test was used as appropriate to compare proportions for categorical variables. Logistic regression analysis was used to identify predictors of pulmonary embolism and its survival. A confidence interval of 95% (*p* < 0.05) was applied to represent the statistical significance of the results, and the level of significance was assigned as 5%. SPSS (Statistical Package for the Social Sciences) version 25.0 software (IBM, Chicago, IL, United States) was used to perform all statistical analysis.

### 2.8. Ethical Approval

This study was approved by the institutional ethics board at the Ministry of Health in Saudi Arabia (No. H-02-K-076-0920-386).

## 3. Results

### 3.1. Patient’s Baseline Characteristics

A total of 159 patients participated in the study, of which 51 were the cases (patients with pulmonary embolism) and 108 patients formed the control group (patients without pulmonary embolism). The incidence of PE among hospitalized patients was around 32%. The case and control group showed comparable baseline characteristics; there was no statistically significant difference between the two groups in term of the gender distribution, duration of stay at the hospital, disease history, and most of the reported signs and symptoms (*p* > 0.05). For further details on the baseline characteristics, refer to Table 1 below.

### 3.2. Vital Signs and Blood Test Findings

The case and control groups were comparable in terms of their vital sign measurements (respiratory rate, heart rate, oxygen saturation) and various blood test measurements (hematocrit, HGB, WBC, INR, and PTT). However, PT and D-dimer measures were significantly different between the two groups, where the mean PT value was higher in the control group compared to the case group, and the median D-dimer value was higher among the case group compared to the control group. For further details on vital signs and blood test findings, refer to Table 2.

### 3.3. Patient’s Radiological Findings

The majority of pulmonary emboli were located in the segmental branch. There was no statistically significant difference between the case and the control groups in terms of the CT parenchymal findings (*p* > 0.05). RT ventricular dysfunction was more common among the case group compared to the control group, whereas the distribution of LT ventricular dysfunction and systolic pulmonary hypertension between the two groups was the same (Table 3).

### 3.4. Risk Factors of Pulmonary Embolism and Its Survival

Using unadjusted logistic regression, we identified that age, smoking history, a higher respiratory rate, and a higher D-dimer value were associated with a higher risk of developing PE (*p* < 0.05). A higher respiratory rate, higher WBC, higher INR value, longer duration between admission and spiral, longer duration of stay at the hospital, requiring ICU admission, duration of stay at the ICU, and requiring intubation were associated with higher odds of death and a decrease in the possibility of survival among hospitalized patients with PE (*p* < 0.05). On the other hand, a higher HGB value was associated with higher odds of survival (*p* < 0.05).

Adjusting for relevant confounding variables such as age, gender, BMI, renal failure, malignancy, hematological diseases, duration between admission and spiral, duration in the ICU, and intubation, we identified that smoking history, a low level of oxygen saturation, and higher D-dimer values were important risk factors that were associated with a higher risk of developing PE (*p* < 0.05). A higher respiratory rate was associated with higher odds of death and a decrease in the possibility of survival among hospitalized patients with PE. For further details on the findings of the logistic regression analysis, refer to Table 4.

## 4. Discussion

In this case-control study, we investigated the incidence, risk factors, and clinical outcomes among hospitalized patients with pulmonary embolism and COVID-19. The key findings of this study are that smoking history, a low level of oxygen saturation, and higher D-dimer values were important risk factors that were associated with a higher risk of developing PE (*p* < 0.05). A higher respiratory rate was associated with higher odds of death and a decrease in the possibility of survival among hospitalized patients with PE. We found that around 32% of hospitalized patients with COVID-19 who underwent CTPA were diagnosed with PE. This finding coincides with previous studies, which reported that about 37.1% of COVID-19 patients were diagnosed with PE [2]. Patients with smoking history values were associated with a higher risk of developing PE. Previous reviews reported a higher risk in current smokers than in individuals who have never smoked [3]. Smoking may promote VTE via different mechanisms such as a procoagulant state, reduced fibrinolysis, inflammation, and increased blood viscosity [4,5].

In our study, elevated D-dimer was found in all patients with PE and non-PE; an elevated D-dimer was found in other reports on COVID-19 associated with pneumonia. Researchers attributed this elevation to excessive consumption of coagulation factors. Some other studies showed that up to 90% of patients admitted to the hospital for pneumonia had high procoagulant markers, with D-dimer being one of the most common [6]. In our study, there is a significant elevation of D-dimer in the PE group compared to the control group, indicating that CTPA is necessary for the exclusion of PE if there is clinical deterioration. This finding coincides with previous studies, which reported that elevated D-dimer levels correlate with the presence of VTE in COVID-19 patients [7,8]. In our study, patients without PE had a median D-dimer level of 3.7, similar to 3.3 observed in patients with COVID-19 in the study by Reich et al. [9].

Pharmacological thromboprophylaxis has shown promise in preventing venous thromboembolism (VTE) for high-risk individuals. The incidence of VTE ranges from 5% to 15% and can be effectively reduced by one-half to two-thirds with appropriate thromboprophylaxis [10]. In critically ill patients, the incidence of deep vein thrombosis (DVT) ranges from 13% to 31% without thromboprophylaxis [11]. However, this risk can be reduced with pharmacological thromboprophylaxis [12]. In our study, all patients were on thromboprophylaxis per guidelines during hospitalization. This finding heightened the risk of venous thromboembolism in COVID-19, despite prophylactic anticoagulation.

The association between COVID-19 and hypercoagulopathy had been established in the literature. This hypercoagulable state is found to be correlated with the severity of the illness since both the incidence and the coagulation abnormalities are more apparent in severe cases. Furthermore, the presence of coagulation abnormalities such as elevated D-dimer has been linked with both mortality and the need for mechanical ventilation [13]. Despite receiving anticoagulation with at least one prophylactic dose, the incidence of VTE was approximately 24% [8]. Several mechanisms contributed to the pathogenesis of micro and macro thrombotic complications in COVID-19. These include the systemic activation of coagulation, and, like other viruses (SARS-CoV-1 and MERS-CoV), SARS-CoV-2 promotes endothelial dysfunction, vascular leak, and pulmonary microthrombi [14,15,16].

The major entry receptors of SARS-CoV-2, the angiotensin converting enzyme 2 (ACE2), are located at the cell surfaces of many organs and regulate renin angiotensin system (RAS). This binding results in dysregulation of RAS and an increase in the pro-inflammatory cytokines ensuing endothelial dysfunction and micro thrombosis [17,18,19]. Autopsy studies of COVID-19 cases have shown that venous thromboembolic events (PE or DVT) are common. Additionally, features of micro thrombosis and endotheliitis were reported not only in the lungs [20,21], but also in the heart, intestine, and kidneys [22]. Additional mechanisms were also described as antiphospholipid antibodies (APSA), where recent publications have reported the association of APSA with thrombotic complications observed in COVID-19 patients [23].

We did not find significant differences between the PE and non-PE group regarding ICU admission, connection to mechanical ventilation, survival, or duration of hospital stay. However, the mortality in patients with PE was around 25.5%; this finding coincides with previous studies, which reported a mortality rate of about 20% [24]. In addition, we found no differences between both groups in the other risk factors for PE, in the duration of hospital admission, and in the duration between admission and the time of CTPA between both groups; however, this may be due to the up-regulation of procoagulant activity in COVID-19 infection, increasing the risk of PE. The coexistence of pneumonia and PE has been known for years, and data from the international cohort RIETE showed that patients with respiratory infections had a higher risk of PE procoagulant activity than patients with other types of infections [25]. Patients with coronavirus disease 2019 (COVID-19) pneumonia may be predisposed to thrombotic complications due to excess inflammatory response, endothelial dysfunction, platelet activation, and stasis of blood flow [26]. In our study, there was no significant difference between both groups in the duration of hospital admission or the duration between admission and the time of CTPA, suggesting that there are other mechanisms related to COVID-19 that promote the coagulopathy, which should be studied in the future.

The results of this study demonstrated a high incidence of pulmonary embolism among hospitalized patients with COVID-19, and this finding should alert health care providers to suspect a diagnosis of PE in COVID-19 patients, especially when clinical signs are associated with significant elevation of D-dimer. Further studies are required to determine the duration of thromboprophylaxis in COVID-19 patients.

This study has some limitations. First, the number of patients included in the study was small, and therefore, we were not able to match patients. Second, the study population only included patients from a single-center hospital in Saudi Arabia. Third, due to the retrospective nature of the study, we did not have data on the CTPA for all suspected PE patients who were limited by clinical instability or a refusal to do CT with IV contrast. In this study, we did not have data for the history of VTE, sepsis, or estrogen use, and we were not able to adjust for these variables. However, we adjusted for multiple other factors. In addition, we were not able to determine if the PE occurred in hospital or before hospital admission.

## 5. Conclusions

Our results revealed a high incidence of pulmonary embolism among hospitalized patients with COVID-19. Preventive measures should be considered for hospitalized patients with smoking history, low level of oxygen saturation, high D-dimer values, and a high respiratory rate. More studies are required to evaluate the risk factors and the mechanisms of thrombosis related to COVID-19.

## Figures and Tables

**Table 1 ijerph-18-07645-t001:** Baseline characteristics for the study sample.

Demographic Variable	Patients with Pulmonary Embolism (Case) (*n* = 51)	Patients without Pulmonary Embolism (Control) (*n* = 108)	*p*-Value
Demographics			
Age, years	56.9 (12.3)	50.9 (15.2)	0.009 **
Gender, No. (%)	40 (78.4)	71 (65.7%)	0.139
Male	40 (78.4)	71 (65.7%)	0.139
BMI, kg/m^2^	27.7 (5.8)	27.9 (5.8)	0.609
Smoking history			
Smoker	32 (62.7)	29 (26.9)	0.000 ***
Comorbidities, No. (%)			
Diabetes mellitus	24 (47.1)	43 (39.8)	0.388
Hypertension	21 (41.2)	38 (35.2)	0.465
Ischemic heart disease	10 (19.6)	17 (15.7)	0.544
Heart failure	3 (5.9)	6 (5.6)	0.934
Renal failure	0	7 (6.5)	0.060
Malignancy	1 (2.0)	0	0.144
HIV	0	1 (0.9)	0.491
Rheumatologic disease, No. (%)			
Rheumatoid arthritis	0	1 (0.9)	0.491
Antiphospholipid syndrome	0	1 (0.9)	0.491
Haematological disease, No. (%)			
Sickle cell disease	0	3 (2.8)	0.230
Pulmonary disease (other than pulmonary embolism), No. (%)			
COPD	1 (2.0)	7 (6.5)	0.854
Asthma	1 (2.0)	8 (7.4)	0.719
Tuberculosis	0	2 (1.9)	0.544
IPF	0	1 (0.9)	0.676
Pulmonary hypertension	1 (2.0)	0	0.012 *
Sign and symptoms (at presentation to hospital), No. (%)			
Fever	44 (86.3)	80 (74.1)	0.083
Cough	41 (80.4)	85 (78.7)	0.806
Sore throat	20 (39.2)	47 (43.5)	0.608
Dyspnoea	47 (92.2)	98 (90.7)	0.769
Haemoptysis	7 (13.5)	11 (10.2)	0.511
Chest pain	21 (41.2)	26 (24.1)	0.027 *
Vomiting	6 (11.8)	28 (25.9)	0.042 *
Diarrhea	15 (29.4)	33 (30.6)	0.883
Nausea	21 (41.2)	39 (36.1)	0.539
Loss of smell	4 (7.8)	18 (16.7)	0.133
Loss of taste	4 (7.8)	17 (15.7)	0.170
Headache	22 (43.1)	42 (38.9)	0.610
Bone ache	29 (56.9)	64 (59.3)	0.775
Other, No. (%)			
Duration of stay at hospital, median days (IQR)	13.00 (13.00)	15.00 (11.00)	0.983
Duration between admission and spiral, median days (IQR)	6.00 (7.00)	5.00 (6.75)	0.953
Outcomes, No. (%)			
Survived	38 (74.5)	74 (68.5)	0.440
Died	13 (25.5)	34 (31.5)	0.440
ICU admission (yes)	39 (76.5)	80 (74.1)	0.745
Intubation (yes)	60 (55.6)	32 (62.7)	0.391
Duration of stay at the ICU, median days (IQR)	6.00 (10.00)	7.00 (13.00)	0.524
Medications and Management, No. (%)			
Heparin	47 (92.2)	1 (0.9)	0.000 ***
Mechanical thrombectomy	3 (5.9)	0	0.803
Alteplase	1 (2.0)	0	0.888
Complications of anticoagulant, No. (%)			
Minor bleeding	3 (5.9)	0	0.011 *

COPD: Chronic obstructive pulmonary diseases; IPF: Idiopathic pulmonary fibrosis; ICU: Intensive care unit; IQR: Interquartile range. * *p* < 0.05, ** *p* < 0.01, *** *p* < 0.001.

**Table 2 ijerph-18-07645-t002:** Vital signs and blood tests upon admission.

Variable.	Patients with Pulmonary Embolism (Case)	Patients without Pulmonary Embolism (Control)	*p*-Value
Respiratory rate	30.5 (5.1)	28.3 (6.3)	0.717
Heart rate	113.9 (13.9)	108.9 (20.5)	0.686
Oxygen saturation	83.7 (8.3)	85.8 (6.5)	0.455
Haematocrit	0.38 (0.07)	0.75 (3.6)	0.204
HGB	127.7 (22.9)	122.0 (24.8)	0.913
WBC, median (IQR)	10.27 (7.06)	9.44 (7.00)	0.188
Platelet count, median (IQR)	251.00 (135.00)	256.00 (144.0)	0.971
INR	1.09 (0.17)	1.13 (0.28)	0.938
PTT, median (IQR)	35.30 (8.30)	34.30 (7.40)	0.324
PT	13.3 (1.73)	14.50 (4.70)	0.044 *
D-dimer, median (IQR)	13.70 (14.38)	3.70 (4.53)	0.000 ***

HGB: Hemoglobin; INR: International normalized ratio; IQR: Interquartile range; PTT: Partial thromboplastin time; PT: Prothrombin time; WBC: White blood cells. Hemoglobin (g/L), Hematocrit (L/L), WBC (10^9^/L), Platelet (10^9^/L), PTT (sec), PT (sec), D-dimer (mg/L). * *p* <0.05, *** *p* < 0.001.

**Table 3 ijerph-18-07645-t003:** Radiological and ECHO findings.

Variable	Patients with Pulmonary Embolism (Case)	Patients without Pulmonary Embolism (Control)	*p*-Value
Location of pulmonary embolism			
Segmental branch	23 (45.1)	-	-
Main pulmonary trunk	12 (23.5)	-	-
Lobar branch	10 (19.6)	-	-
Sub segmental	6 (11.8)	-	-
CT parenchymal findings			
Bilateral peripheral ground glass	26 (51.0)	50 (46.3)	0.581
Bilateral peripheral ground glass with consolidation	23 (45.1)	50 (46.3)	0.887
Unilateral peripheral ground glass	2 (3.9)	8 (7.4)	0.398
ECHO			
RT ventricular dysfunction	25 (49.0)	22 (20.4)	0.000 ***
LT ventricular dysfunction	4 (7.8)	8 (7.4)	0.575
Systolic pulmonary hypertension	14 (27.5)	24 (22.2)	0.298

*** *p* < 0.001.

**Table 4 ijerph-18-07645-t004:** Risk factors of pulmonary embolism and its survival.

Variable	Risk Factors of Pulmonary Embolism				Factors Affecting Survival among Patients with Pulmonary Embolism			
	OR (95% CI)	*p*-Value	AOR ^§^ (95% CI)	*p*-Value	OR (95% CI)	*p*-Value	AOR ^§^ (95% CI)	*p*-Value
Demographics and social								
Age (years)	1.03 (1.01–1.06)	0.018 *	-	-	0.99 (0.96–1.01)	0.230	-	-
Gender (female)	0.53 (0.24–1.15)	0.107	-	-	1.19 (0.56–2.53)	0.653	-	-
Smoking (yes)	4.59 (2.26–9.33)	0.000 ***	3.85 (1.73–8.58)	0.001 **	1.00 (0.50–2.02)	0.991	1.70 (0.46–6.27)	0.422
Obesity (BMI > 30 kg/m^2^)	1.40 (0.70–2.81)	0.337	-	-	0.87 (0.43–1.78)	0.703	-	-
Comorbidities								
Diabetes mellitus (yes)	1.34 (0.69–2.63)	0.388	1.02 (0.47–2.19)	0.969	0.76 (0.38–1.52)	0.440	1.14 (0.35–3.71)	0.833
Hypertension (yes)	1.29 (0.65–2.55)	0.466	1.21 (0.56–2.61)	0.623	0.64 (0.32–1.28)	0.202	0.65 (0.20–2.09)	0.473
Ischemic heart disease (yes)	1.31 (0.55–3.10)	0.545	1.05 (0.40–2.77)	0.915	0.81 (0.33–1.96)	0.638	0.88 (0.20–3.87)	0.870
Heart failure (yes)	1.06 (0.26–4.43)	0.934	1.22 (0.25–6.09)	0.806	0.50 (0.13–1.96)	0.322	0.05 (0.00–1.24)	0.068
Renal failure (yes)	-	-	-	-	-	-	-	
Malignancy (yes)	-	-	-	-	-	-	-	
Rheumatologically disease (yes)	-	-	-	-	-	-	-	
Haematological disease (yes)	-	-	-	-	-	-	-	
COPD (yes)	0.79 (0.06–10.38)	0.855	-	-	0.90 (0.12–7.03)	0.920	-	
Asthma (yes)	0.63 (0.05–8.20)	0.720	-	-	1.17 (0.15–9.01)	0.882	-	
Tuberculosis (yes)	-	-	-	-	-	-	-	
IPF (yes)	-	-	-	-	-	-	-	
Pulmonary hypertension (yes)	-	-	-	-	-	-	-	
Vital signs								
Respiratory rate	1.07 (1.00–1.15)	0.043 *	1.07 (0.99–1.15)	0.076	0.91 (0.84–0.98)	0.011 *	0.82 (0.71–0.94)	0.004 **
Heart rate	1.02 (1.00–1.04)	0.067	1.02 (1.00–1.05)	0.061	0.98 (0.96–1.00)	0.113	0.97 (0.94–1.01)	0.165
Oxygen saturation	0.98 (0.95–1.02)	0.279	0.95 (0.90–1.00)	0.039 *	1.02 (0.99–1.06)	0.272	1.05 (0.97–1.15)	0.230
Laboratory								
Haematocrit	0.89 (0.48–1.64)	0.705	0.74 (0.01–65.73)	0.895	-	-	-	-
HGB	1.01 (0.99–1.02)	0.438	1.00 (0.98–1.02)	0.790	1.03 (1.01–1.04)	0.001 **	1.01 (0.98–1.03)	0.528
WBC	1.02 (0.97–1.07)	0.558	1.02 (0.97–1.08)	0.407	0.94 (0.89–1.00)	0.035 *	0.97 (0.88–1.06)	0.468
Platelet count	1.00 (1.00–1.00)	0.615	1.00 (1.00–1.00)	0.972	1.00 (1.00–1.00)	0.580	1.00 (1.00–1.00)	0.819
PT	0.91 (0.80–1.04)	0.174	0.91 (0.80–1.04)	0.180	1.04 (0.94–1.16)	0.434	1.00 (0.813–1.23)	0.985
PTT	1.00 (0.99–1.01)	0.642	1.00 (0.99–1.01)	0.491	1.00 (1.00–1.01)	0.645	1.03 (0.97–1.09)	0.310
INR	1.07 (0.26–4.34)	0.925	2.31 (0.30–17.65)	0.419	0.12 (0.02–0.69)	0.018 *	0.05 (0.00–1.20)	0.065
D dimer	1.17 (1.10–1.24)	0.000 ***	1.21 (1.13–1.30)	0.000 ***	0.99 (0.96–1.01)	0.356	0.99 (0.95–1.03)	0.685
Other								
Duration between admission and spiral (days)	0.99 (0.92–1.07)	0.862	-	-	0.89 (0.83–0.96)	0.003 **	-	-
Severity of disease (required ICU admission)	1.14 (0.52–2.47)	0.745	1.27 (0.46–3.48)	0.646	0.04 (0.01–0.31)	0.002 **	1.79 (0.09–36.12)	0.705
Duration of hospital stay (days)	1.00 (0.96–1.04)	0.907	1.01 (0.96–1.07)	0.695	0.96 (0.93–1.00)	0.042 *	1.10 (1.00–1.21)	0.053
Duration of stay in the ICU (days)	0.98 (0.94–1.02)	0.395	-	-	0.86 (0.82–0.91)	0.000 ***	-	-
Intubation (yes)	0.74 (0.38–1.47)	0.392	-	-	0.01 (0.00–0.05)	0.000 ***	-	-

BMI: Body mass index; COPD: Chronic obstructive pulmonary diseases; ICU: Intensive care unit; IQR: Interquartile range; HGB: Hemoglobin: International normalized ratio; IPF: Idiopathic pulmonary fibrosis; Prothrombin time; PTT: Partial thromboplastin time; PT: Prothrombin time; WBC: White blood cells. ^§^ Adjusted for the following variables: age, gender, BMI, renal failure, malignancy, hematological diseases, duration between admission and spiral, duration in the ICU, and intubation. * *p* < 0.05, ** *p* < 0.01, *** *p* < 0.001.

## Data Availability

The data that support the findings of this study are available from the corresponding author, upon reasonable request.

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
