# Peer review of "Incidence and Outcomes of Pulmonary Embolism among Hospitalized COVID-19 Patients"

_ijerph, 2021, doi:10.3390/ijerph18147645_

Round 1

Reviewer 1 Report

Greetings authors,

 In this manuscript, the authors addressed the Incidence and Outcomes of Pulmonary embolism among hospitalized COVID-19 patients. The article is reasonably well written. Here are some recommendations. 

  1. Since the authors mentioned that most of the baseline characteristics are similar, propensity scoring if possible would add more value to it. 
  2. Consider adding "History of VTE and estrogen " in baseline characteristics. 
  3. Outcomes: In terms of the outcomes, there would be several factors other than PE, like sepsis, that might be contributing to the results, including mortality, LOS, etc. It would be better to adjust for sepsis, pressors, acute organ failure(liver/kidney) apart from their baseline chronic disease. 
  4. The factors mentioned above should be controlled so that we can accept the variables mentioned by the authors can contribute to poor outcomes in COVID patients with PE. 

Minor: 

Typo in line 147: systolic pulmonary embolism instead of pulmonary hypertension

Author Response

Manuscript ID; ijerph-1257056
Title: Incidence and Outcomes of Pulmonary Embolism Among Hospitalized COVID-19 patients

Corresponding Author: Dr. Fatemah M Alsaleh

Dear Editor,

Thank you for the opportunity to revise and resubmit our manuscript based on the reviewers’ comments. Please find below our itemized point-by-point responses to the journal requirements and reviewers’ comments. Answers are written in blue font.

Reviewer 1 
In this manuscript, the authors addressed the Incidence and Outcomes of Pulmonary embolism among hospitalized COVID-19 patients. The article is reasonably well written. Here are some recommendations.

Since the authors mentioned that most of the baseline characteristics are similar, propensity scoring if possible would add more value to it.

We thank the reviewer for the comments. Propensity is a powerful statistical method, and it is widely used in observational cohort studies to address the confounding issues (1). However, our study sample consisted only of 159 patients, and therefore, using propensity score matching or weighting will decrease the sample size significantly and decrease the precision of our estimates. Additionally, as the reviewer mentioned “the baseline characteristics are similar for most of the baseline variables”, which increase the reliability of our comparison between the case and the control group. Therefore, we were unable to use PS in our study.

Consider adding "History of VTE and estrogen " in baseline characteristics.

History of VTE and eestrogen use are important risk factors in the development of PE or DVT. However, unfortunately, this data was not available from the data extracted. We have now mentioned this in the limitation of the study, please refer to page 17-18, lines number 250-256.

Outcomes: In terms of the outcomes, there would be several factors other than PE, like sepsis, that might be contributing to the results, including mortality, LOS, etc. It would be better to adjust for sepsis, pressors, acute organ failure(liver/kidney) apart from their baseline chronic disease.

The factors mentioned above should be controlled so that we can accept the variables mentioned by the authors can contribute to poor outcomes in COVID patients with PE.

We thank the reviewer for these valuable comments, and we agree that these factors are important factors. However, as mentioned above we did not have these data and we have now mentioned this in the limitation of the study, please refer to pages 17-18, lines number 250-256.

Minor:

Typo in line 147: systolic pulmonary embolism instead of pulmonary hypertension
We have now addressed this in the main manuscript. Please refer to the results section, page number 10 lines number 158.

Reviewer 2 Report

The authors have conduced a retrospective case-control study to investigate the incidence of pulmonary embolism among hospitalized patients with COVID-, risk factors, and the impact on survival.

I would congratulate with the Authors for the excellent work. Besides, I would suggest some minor improvements:

Please correct: Patients with a confirmed COVID-19 diagnosis by a real-time polymerase chain reaction (PCR) and without with a confirmed diagnosis of pulmonary embolism formed the control case group.

Please, in Table 1: correct male alone with gender (male); insert a Table legend

Table 2: please insert the unit of measure for each variable and the table legend

Table 4: same as Table 2

Author Response

Manuscript ID; ijerph-1257056
Title: Incidence and Outcomes of Pulmonary Embolism Among Hospitalized COVID-19 patients

Corresponding Author: Dr. Fatemah M Alsaleh

Dear Editor,

Thank you for the opportunity to revise and resubmit our manuscript based on the reviewers’ comments. Please find below our itemized point-by-point responses to the journal requirements and reviewers’ comments. Answers are written in blue font.

Reviewer 2

The authors have conduced a retrospective case-control study to investigate the incidence of pulmonary embolism among hospitalized patients with COVID-, risk factors, and the impact on survival. I would congratulate with the Authors for the excellent work. Besides, I would suggest some minor improvements:

Please correct: Patients with a confirmed COVID-19 diagnosis by a real-time polymerase chain reaction (PCR) and without with a confirmed diagnosis of pulmonary embolism formed the control case group.

Thank you for your comment. We have now corrected this in the main manuscript. Please refer to the methods section (case definition), page 4, line number 108.

Please, in Table 1: correct male alone with gender (male); insert a Table legend

We have now corrected this in the main manuscript. Please refer to the results section, table 1, page 6.

Table 2: please insert the unit of measure for each variable and the table legend

We have now corrected this in the main manuscript. Please refer to the results section, table 2, page 9.

Table 4: same as Table 2

Thank you for your comments. We have now corrected this in the main manuscript. Please refer to the results section, table 4, page 13.

Reviewer 3 Report

Authors have analyzed the incidence of pulmonary embolism among hospitalized COVID-19 patients as well as predictors of pulmonary embolism and its effect on survival in Middle East population. Similar studies have been performed using different populations including, arabia Saudi population (Alharthy A et al. J Epidemiol Glob Health. 2021 Mar;11(1):98-104; Desai R, et al. SN Compr Clin Med. 2020;28:1-4; Martínez Chamorro E, et al. Radiologia (Engl Ed). 2021;63(1):13-21; etc) and even meta-analysis (Ng JJ, et al. J Intensive Care. 2021;9(1):20. Roncon L et al. Eur J Intern Med. 2020 Dec;82:29-37. doi: 10.1016/j.ejim.2020.09.006.). This fact seems to decrease originality of Badr´s study.

There are different aspect which may significantly limit quality of findings reported by Badr and collaborators:

1.- Authors have described that a retrospective case-control study was conducted. However, there are different aspects which should be clarified. Why cases and controls were not matched according to gender and age? Why sample size estimation analysis was not performed? What case/control proportion should be recruited according to pulmonary embolism prevalence reported by other authors? etc. In addition, it is not clear if case subjects were recruited with confirmed diagnosis of pulmonary embolism of it appeared for hospital stay. Inclusion and exclusion criterion subsection should be added. Indeed, after carefully reading the method section, the study design may be interpreted as a retrospective cohort study. In this regard, according to data described by authors, it would be interpreted that 159 COVID-19 patients were recruited between March 15,2020 and June 15,2020, and they were followed up August 15, 2000. In this time period of following were evaluated different outcomes, including apparition of pulmonary embolism among others. This study design would allow determining “incidence” of pulmonary embolism among COVID-19 patients.

2.- Among other, the features of anticoagulant and antiplatelet drugs treatment of patients must be provided, it is crucial to discuss pulmonary embolism incidence.

3.- There are several mistakes or inconsistences in results tables. For example, what is the meaning of *, ** or ***?  Asterisks are not required since P values are provided. In the table 2, there are not measurement units of interest variables (D-dimer, PT,PTT, platelet count, etc.). What reference group was considered for the age variable in the table 4?

4.- Sample size seems to be too limited to obtain a real incidence of pulmonary embolism. Author should justify it.

Author Response

Manuscript ID; ijerph-1257056
Title: Incidence and Outcomes of Pulmonary Embolism Among Hospitalized COVID-19 patients

Corresponding Author: Dr. Fatemah M Alsaleh

Dear Editor,

Thank you for the opportunity to revise and resubmit our manuscript based on the reviewers’ comments. Please find below our itemized point-by-point responses to the journal requirements and reviewers’ comments. Answers are written in blue font.

Reviewer 3

Authors have analyzed the incidence of pulmonary embolism among hospitalized COVID-19 patients as well as predictors of pulmonary embolism and its effect on survival in Middle East population. Similar studies have been performed using different populations including, Arabia Saudi population (Alharthy A et al. J Epidemiol Glob Health. 2021 Mar;11(1):98-104; Desai R, et al. SN Compr Clin Med. 2020;28:1-4; Martínez Chamorro E, et al. Radiologia (Engl Ed). 2021;63(1):13-21; etc) and even meta-analysis (Ng JJ, et al. J Intensive Care. 2021;9(1):20. Roncon L et al. Eur J Intern Med. 2020 Dec;82:29-37. Doi: 10.1016/j.ejim.2020.09.006.). This fact seems to decrease originality of Badr´s study.

We thank the reviewer for this constructive comment. We also recognise that there are previous studies that investigated the association and incidence of PE among patients with COVID-19 (2), including a study by Alharthi et al that was conducted in Saudi Arabia (3). However, unlike the previous study that was conducted in Saudi Arabia, the aim of our study was to assess the incidence of pulmonary embolism among hospitalized patients with COVID-19, risk factors. We also used a different study design which is also superior to the study design used in the former study, as they used a cross sectional study design. In addition, the previous study was conducted in a different region in Saudi Arabia, and therefore, given the emergent nature of this pandemic we assume that providing the literature with more data from different populations and geographical location will add important value to the literature.

There are different aspect which may significantly limit quality of findings reported by Badr and collaborators:

1.- Authors have described that a retrospective case-control study was conducted. However, there are different aspects which should be clarified.

Why cases and controls were not matched according to gender and age?

We used all available data at Al-Noor Specialist Hospital in Mecca, Saudi Arabia. All patients who were admitted during the study period and met the inclusion criteria were included. Due to the small sample size in our study, we were not able to match the two groups “case and control groups” based on the age. However, as you can see there was no statistically significant difference between the two groups based on the age (p=0.139). We have now highlighted this point in the limitation section, please refer to page 17-18, lines number 68-74.

Why sample size estimation analysis was not performed?

Thank you for your comment. The reason why we did report any sample size calculation, is that there are no prior studies that have been conducted in Saudi Arabia in terms of the incidence/prevalence of pulmonary embolism among patients hospitalised with COVID-19. This limited our ability to do sample size calculation. Additionally, we included all patients who were admitted during the study period and met the inclusion criteria in our study and the maximum achievable number of patients was 159.

What case/control proportion should be recruited according to pulmonary embolism prevalence reported by other authors? Etc.

To the best of our knowledge, we are not aware of any study that investigated the incidence/prevalence of pulmonary embolism among patients hospitalised with COVID-19 in Saudi Arabia.

In addition, it is not clear if case subjects were recruited with confirmed diagnosis of pulmonary embolism of it appeared for hospital stay.

We thank the reviewer for this comment as the prolonged hospital stay is very important risk factor for the development of pulmonary embolism, however, the median duration between admission and development of pulmonary embolism in our study was 6 days (less than 2 weeks) with no significant difference between both groups. Therefore, it is unlikely that this will affect the overall conclusion of this study.

Inclusion and exclusion criterion subsection should be added.

Thank you for your comment. We have now included a subsection for the inclusion and exclusion criteria. Please refer to the methods section page 3, lines number 78-85.

Indeed, after carefully reading the method section, the study design may be interpreted as a retrospective cohort study. In this regard, according to data described by authors, it would be interpreted that 159 COVID-19 patients were recruited between March 15,2020 and June 15,2020, and they were followed up August 15, 2000. In this time period of following were evaluated different outcomes, including apparition of pulmonary embolism among others. This study design would allow determining “incidence” of pulmonary embolism among COVID-19 patients.

We agree with the author that cohort study design is a good methodology to address the objectives of this the study, especially that PE is not a rare disease among patients with covid-19 as reported in the literature. However, the reason why we chose a case control design is that we were interested to investigate the risk factors associated with PE, and a case control design will allow us to explore multiple risk factors and unlike cohort design which has the advantage of exploring multiple outcomes.

2.- Among other, the features of anticoagulant and antiplatelet drugs treatment of patients must be provided, it is crucial to discuss pulmonary embolism incidence.

Thanks for the reviewer for these important comments, and we agree that features of anticoagulants drugs are important. All PE cases were treated with heparin alone, thrombolytic (alteplase) or mechanical thrombectomy, however, antiplatelet drugs were not used as a part of treatment of pulmonary embolism. Please see table 1 below. In addition, we have now corrected this in the main manuscript in the results section in table 1, page 6.

Demographic variable

Patients with Pulmonary embolism (case) (n= 51)

Patients without Pulmonary embolism (control) (n= 108)

P-Value

Demographics

Age (years)

56.9 (12.3)

50.9 (15.2)

0.009**

Gender (male) No. (%)

40 (78.4)

71 (65.7%)

0.139

BMI (kg/m2)

27.7 (5.8)

27.9 (5.8)

0.609

Smoking history

Smoker

32 (62.7)

29 (26.9)

0.000***

Comorbidities No. (%)

Diabetes mellitus

24 (47.1)

43 (39.8)

0.388

Hypertension

21 (41.2)

38 (35.2)

0.465

Ischemic heart disease

10 (19.6)

17 (15.7)

0.544

Heart failure

3 (5.9)

6 (5.6)

0.934

Renal failure

0

7 (6.5)

0.060

Malignancy

1 (2.0)

0

0.144

HIV

0

1 (0.9)

0.491

Rheumatologically disease No. (%)

Rheumatoid arthritis

0

1 (0.9)

0.491

Antiphospholipid syndrome

0

1 (0.9)

0.491

Haematological disease No. (%)

Sickle cell disease

0

3 (2.8)

0.230

Pulmonary disease (other than pulmonary embolism) No. (%)

COPD

1 (2.0)

7 (6.5)

0.854

Asthma

1 (2.0)

8 (7.4)

0.719

Tuberculosis

0

2 (1.9)

0.544

IPF

0

1 (0.9)

0.676

Pulmonary hypertension

1 (2.0)

0

0.012*

Sign and symptoms (at presentation to hospital) No. (%)

Fever

44 (86.3)

80 (74.1)

0.083

Cough

41 (80.4)

85 (78.7)

0.806

Sore throat

20 (39.2)

47 (43.5)

0.608

Dyspnoea

47 (92.2)

98 (90.7)

0.769

Haemoptysis

7 (13.5)

11 (10.2)

0.511

Chest pain

21 (41.2)

26 (24.1)

0.027*

Vomiting

6 (11.8)

28 (25.9)

0.042*

Diarrhea

15 (29.4)

33 (30.6)

0.883

Nausea

21 (41.2)

39 (36.1)

0.539

Loss of smell

4 (7.8)

18 (16.7)

0.133

Loss of taste

4 (7.8)

17 (15.7)

0.170

Headache

22 (43.1)

42 (38.9)

0.610

Bone ache

29 (56.9)

64 (59.3)

0.775

Other No. (%)

Duration of stay at hospital (median days (IQR))

13.00 (13.00)

15.00 (11.00)

0.983

Duration between admission and spiral (median days (IQR))

6.00 (7.00)

5.00 (6.75)

0.953

Outcomes No. (%)

Survived

38 (74.5)

74 (68.5)

0.440

Died

13 (25.5)

34 (31.5)

0.440

ICU admission (yes)

39 (76.5)

80 (74.1)

0.745

Intubation (yes)

60 (55.6)

32 (62.7)

0.391

Duration of stay at the ICU (median days (IQR))

6.00 (10.00)

7.00 (13.00)

0.524

Medications and management No. (%)

Heparin

47 (92.2)

1 (0.9)

0.000***

Mechanical thrombectomy

3 (5.9)

0

0.803

Alteplase

1 (2.0)

0

0.888

Complications of anticoagulant No. (%)

Minor bleeding

3 (5.9)

0

0.011*

3.- There are several mistakes or inconsistences in results tables. For example, what is the meaning of *, ** or ***? 

These stars stands for the degree of significance as the following (*p<0.05,  **p<0.01, ***p<0.001).

Asterisks are not required since P values are provided.

- We added them to highlight it further and make it clearer to the reader.

 In the table 2, there are not measurement units of interest variables (D-dimer, PT,PTT, platelet count, etc.).

We thank the reviewer for raising our attention to this point. Please see the details below in table 2. In addition, we have now added these measurement in the main manuscript in the results section in table 2.

Item

Reference

Haemoglobin

120 – 150          g/L

Haematocrit

0.36 – 0.46        L/L

WBC

4 – 11                109 / L

Platlet

150 – 400          109 / L

PTT

26 – 39               sec

INR

0.8 – 1.2

Prothrombin time PT

11 – 16               sec

D-dimer

0 – 0.55             mg/L

What reference group was considered for the age variable in the table 4?

The age was handled as numeric variable in the regression model, therefore, there is no reference age group for it.

4.- Sample size seems to be too limited to obtain a real incidence of pulmonary embolism. Author should justify it.

We used all available data at Al-Noor Specialist Hospital in Mecca, Saudi Arabia. All patients who were admitted during the study period and met the inclusion criteria were included. We have now highlighted this point in the limitation section.

  1. Austin PC. An Introduction to Propensity Score Methods for Reducing the Effects of Confounding in Observational Studies. Multivariate Behav Res. 2011;46(3):399-424.
  2. Jevnikar M, Sanchez O, Chocron R, Andronikof M, Raphael M, Meyrignac O, et al. Prevalence of pulmonary embolism in patients with COVID 19 at the time of hospital admission. Eur Respir J. 2021.
  3. Alharthy A, Aletreby W, Faqihi F, Balhamar A, Alaklobi F, Alanezi K, et al. Clinical Characteristics and Predictors of 28-Day Mortality in 352 Critically Ill Patients with COVID-19: A Retrospective Study. J Epidemiol Glob Health. 2021;11(1):98-104.

Round 2

Reviewer 1 Report

The authors addressed my concerns in the limitations. 

Reviewer 3 Report

Authors have justified in detail all comments of the previous revision. Additionally, they have removed several inconsistences and improved description of studied population. Modified limitation section is welcome.